# Daily-Omni: Towards Audio-Visual Reasoning with Temporal Alignment across Modalities

## Abstract

Recent Multimodal Large Language Models (MLLMs) achieve promising performance on visual and audio benchmarks independently. However, the ability of these models to process cross-modal information synchronously remains largely unexplored. In this paper, we introduce: 1) **Daily-Omni**, an Audio-Visual Questioning and Answering benchmark comprising 684 videos of daily life scenarios from diverse sources, rich in both audio and visual information, and featuring 1197 multiple-choice QA pairs across 6 major tasks; 2) **Daily-Omni QA Generation Pipeline**, which includes automatic annotation, QA generation and QA optimization, significantly improves efficiency for human evaluation and scalability of the benchmark; 3) **Daily-Omni-Agent**, a training-free agent utilizing open-source Visual Language Model (VLM), Audio Language Model (ALM) and Automatic Speech Recognition (ASR) model to establish a baseline for this benchmark. The results show that current MLLMs still struggle significantly with tasks requiring audio-visual integration, but combining VLMs and ALMs with simple **temporal alignment techniques** can achieve substantially better performance.

## 1 Introduction

Non-textual modalities, such as vision and hearing, often convey richer and more nuanced information than text alone in daily life scenarios, playing a crucial role in our understanding of and interaction with the physical world. Therefore, advancements in Multimodal Large Language Models (MLLMs) capable of comprehensively understanding the multi-modal information are a crucial foundation for achieving artificial intelligence capable of interacting with the physical world and observing the outcome of its operations.

Recent MLLMs (Xu et al., 2025; Sun et al., 2024; Pichai et al., 2024; Zhang et al., 2024a; Fu et al., 2025b; Liu et al., 2025c; Cheng et al., 2024; Li et al., 2025; Xie & Wu, 2024; Guo et al., 2025; Team et al., 2024; Microsoft et al., 2025; Lu et al., 2024b; Liu et al., 2025a; Team et al., 2025) have demonstrated groundbreaking capabilities spanning audio (ASR, sound classification, captioning) and visual domains (OCR, VQA, video grounding), with significant accuracy improvements over previous benchmarks. However, existing methods still face several limitations. Firstly, many MLLMs predominantly focus on visual abilities, often neglecting the importance of other modalities like audio. This oversight may stem from the fact that current visual datasets are more abundant, of higher quality, and cover a broader range of tasks compared to audio datasets. Existing audio datasets (Panayotov et al., 2015; Wang et al., 2021; Poria et al., 2019; Chen et al., 2020; Gemmeke et al., 2017; Gong et al., 2022) tend to prioritize speech-related or music-related tasks and basic sound classification, often overlooking more complex yet crucial tasks such as reasoning over generic sounds. Consequently, many MLLMs incorporate only speech encoders as their primary auditory component or rely heavily on speech-related datasets for audio pre-training. This architectural limitation fundamentally restricts their ability to comprehend rich acoustic environments where non-speech sounds (e.g., environmental noises, mechanical failures, or emotional cues in non-verbal vocalizations) carry critical semantic information. Secondly, the current landscape lacks high-quality multimodal datasets that integrate temporally aligned auditory and visual information. Existing audio-visual datasets and benchmarks (Yun et al., 2021; Li et al., 2022; Yang et al., 2022; Li et al., 2024; Hong et al., 2025; Gong et al., 2024a; Sung-Bin et al., 2025; Geng et al., 2025) reveal three persistent limitations. First, several focus on specialized scenarios (Yun et al., 2021; Li et al., 2022) such as musical performances or panoramic environments, thereby introducing domain-specific biases. Second, many employ static image-

Figure 1: **Examples of Daily-Omni QAs.** The audio and visual information required for answering the questions are provided in the figure. The correct answer for the given questions are highlighted. More cases are presented in Appendix A.

audio pairs (Li et al., 2024; Gong et al., 2024a) that disregard crucial temporal dynamics inherent in real-world video contexts. Third, current tasks are often too narrow, with many benchmarks focusing on specific applications such as captioning or open-ended responses (Geng et al., 2025). The common absence of rigorous evaluation frameworks and standardized metrics makes it hard to reliably compare results. While WorldSense (Hong et al., 2025) established a valuable audio-visual multi-choice QA benchmark for daily scenarios through its meticulous dataset curation, two critical limitations persist: (1) it lacks a systematic framework for scalable QA generation, relying instead on manual annotation processes that hinder dataset expansion; and (2) the benchmark serves primarily as an evaluation tool, offering limited methodological guidance on enhancing model capabilities via explicit training protocols or architectural modifications. This dual limitation constrains both the benchmark's adaptability to emerging domains and its practical utility in driving model improvements.

In this paper, we introduce **Daily-Omni**, an Audio-Visual Questioning and Answering benchmark with 684 videos featuring daily life scenarios from various sources (Gemmeke et al., 2017; Fu et al., 2025a; Farré et al., 2024) with rich audio and visual information and 1197 multiple choice QA-pairs across 6 major tasks ranging from audio visual event aligning to complicated cross-modal reasoning. The videos are sampled from different datasets and segmented into 30-second or 60-second intervals to systematically evaluate model performance across different temporal contexts. We further introduce **Daily-Omni QA Generation Pipeline** which encompasses five automated modules: video annotation, annotation revision, audio-visual temporal alignment, QA generation, and QA optimization. This framework demonstrates remarkable scalability: a single annotator can complete quality filtering process for these 1197 high-quality QA pairs within 30 hours, achieving an approximate 30% acceptance rate from initially generated candidates. In addition, we propose **Daily-Omni Agent**, an audio-visual agent utilizing open-source Visual Language Model (VLM), Audio Language Model (ALM) and Automatic Speech Recognition (ASR) model without further finetuning to establish a baseline for this benchmark. We evaluated recent MLLMs and our agent on the Daily-Omni benchmark, where the Daily-Omni Agent achieved **state-of-the-art** performance among open-source methods. The experimental results show that current MLLMs still face significant challenges in tasks requiring deep audio-visual temporal integration. They also reveal that by combining existing visual and audio language models with simple temporal alignment techniques, as demonstrated by our Daily-Omni Agent, substantially improved performance can be achieved, underscoring a promising direction for enhancing multimodal reasoning.

## 2 RELATED WORKS

### 2.1 MULTIMODAL LARGE LANGUAGE MODELS

Recent advancements in Multimodal Large Language Models (MLLMs) primarily fall into three categories: Audio Language Models (ALMs) incorporating audio capabilities (Tang et al., 2024; Gong et al., 2024b; Chu et al., 2023; 2024; Ghosh et al., 2025; 2024; Goel et al., 2025), Visual Language Models (VLMs) adding visual understanding (Liu et al., 2025b; Bai et al., 2025; Lu et al., 2024a; Wang et al., 2024), and Omni-modal Language Models (OLMs) combining both audio and visual modalities (Pichai et al., 2024; Li et al., 2025; Zhang et al., 2024a; Team et al., 2024; Cheng et al., 2024; Liu et al., 2025c; Microsoft et al., 2025; Fu et al., 2025b; Guo et al., 2025; Xu et al., 2025;

Liu et al., 2025a; Team et al., 2025). These MLLMs often employ a modular architecture, utilizing separate encoders for audio and visual inputs. In the audio domain, Radford et al. (2023); Chen et al. (2023); LI et al. (2024) proposed audio encoders to extract sound representations. Some models Tang et al. (2024); Liu et al. (2025c); Zhang et al. (2024a) even integrate multiple audio encoders specialized for different sound types (e.g., speech, music). Similarly, visual encoders (Li et al., 2023; Dai et al., 2023; Liu et al., 2023; Dehghani et al., 2023) are used to process images and videos. However, this modular approach struggles to capture the crucial temporal correlations inherent in synchronized audio-visual streams. This limitation arises because audio and visual inputs are encoded independently. Although multi-modal positional embedding techniques like TMRoPE (Xu et al., 2025) enhance cross-modal temporal understanding to some extent, effective methods for temporally aligning multimodal data remain relatively scarce. Additionally, in several MLLMs (Microsoft et al., 2025; Fu et al., 2025b; Zhang et al., 2024a) , the audio encoder primarily serves to process user instructions–akin to how the text modality is used–rather than perceiving the environment.

## 2.2 AUDIO-VISUAL UNDERSTANDING DATASETS AND BENCHMARKS

The development of uni-modal datasets and associated tasks for audio and vision has driven advancements in Audio Language Models (ALMs) and Visual Language Models (VLMs). Visual datasets and benchmarks (Li et al., 2024b; Fu et al., 2025a; Wu et al., 2021; Mangalam et al., 2023; Liu et al., 2023; Wu et al., 2024; Zhang et al., 2024b; Yue et al., 2024; 2025; Li et al., 2024a; Fu et al., 2024; Gao et al., 2017; Hu et al., 2025) primarily focus on tasks for static images (OCR, grounding, segmentation, classification, and question-answering) and dynamic videos (captioning, temporal grounding, and understanding), while audio datasets and benchmarks (Ghosh et al., 2025; Gemmeke et al., 2017; Chen et al., 2020; Panayotov et al., 2015; Yang et al., 2024) address speech-related tasks (ASR, emotion recognition, and entity recognition) and non-speech tasks (such as sound classification and audio grounding). However, while attempts at creating audio-visual datasets date back to at least 2021 (Yun et al., 2021), these early efforts often had significant limitations. For example, Music-AVQA (Li et al., 2022) focused specifically on music performance videos, while Pano-AVQA (Yun et al., 2021) centered on panoramic videos. Others, such as AVQA (Yang et al., 2022) and OmniBench (Li et al., 2024), were restricted to short, simple videos or static images. Furthermore, AV-Odyssey (Gong et al., 2024a) heavily emphasized specific audio tasks, such as recognizing timbre and loudness, rather than broader audio-visual understanding. While WorldSense (Hong et al., 2025), a concurrent work, also provides a valuable benchmark for real-world audio-visual question-answering, the efficient scalability of AVQA datasets and the enhancement of OLM abilities still require further exploration.

## 3 DAILY-OMNI

This section details the Daily-Omni framework, which comprises three core components: multi-modal data curation, hierarchical video annotation, and QA synthesis and evaluation. Furthermore, we delineate the evaluation paradigm implemented through the Daily-Omni Agent, which serves as our benchmarking baseline for systematic assessment. Daily-Omni aims to establish a benchmarking framework designed to systematically evaluate MLLMs' ability to perceive and reason in real-life audio-visual scenarios. As illustrated in Figure 2, the Daily-Omni benchmark comprises videos from all **11** YouTube categories. The QAs are meticulously crafted to assess a spectrum of multi-modal capabilities, spanning from basic cross-modal perception to intricate reasoning. All questions are designed to require the integration of audio, visual, and textual information for a correct answer. In total, the Daily-Omni benchmark comprises 684 videos and 1197 QA pairs. Of these QAs, 550 correspond to 60-second videos and 647 correspond to 30-second videos. We compared the features of Daily-Omni with various audio-visual benchmarks in Table 1.

### 3.1 DATA CURATION

Daily-life scenarios offer abundant information across both audio and visual modalities. Within the visual modality, our data curation focuses on selecting videos characterized by **significant temporal dynamics**. Conversely, static scenes such as a vlogger addressing the camera with limited motion offer minimal temporal information for questioning and are usually excluded. Regarding the audio modality, recognizing that prior datasets have concentrated heavily on speech, our benchmark is designed to encompass a **broader range of everyday sounds**, such as music, speech, and other sound

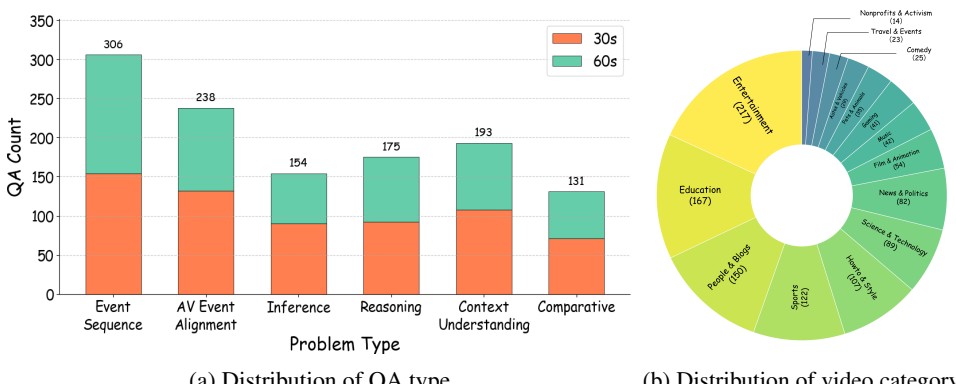

(a) Distribution of QA type  (b) Distribution of video category

Figure 2: **Distribution of 1197 Daily-Omni QA pairs.**

events. These acoustic signals should be present within the videos, appearing either concurrently or consecutively. Furthermore, to maintain focus on audio-visual processing and avoid challenges related to multilingual text comprehension, videos with languages other than English are also excluded.

Table 1: **Comparison of audio-visual benchmarks.** We detail publication, modality (A: audio, V: video, I: image), size, and question type (MC: multiple choice, DF: defined word, BB: bounding boxes). 'Efficient Scalability' refers to automated expansion methods. 'Open-Domain' and 'General Sound' denote genre and sound diversity.

| Benchmarks | Pub | Modality | #Video | #QA Pairs | Question Type | Efficient Scalability | Open-Domain | General Sound |
|---|---|---|---|---|---|---|---|---|
| AVQA | ACM MM'22 | V+A | 57,015 | 57,335 | MC | ✗ | ✓ | ✗ |
| Music-AVQA | CVPR'22 | V+A | 9,288 | 45,867 | DF | ✗ | ✗ | ✓ |
| Pano-AVQA | ICCV'21 | V+A | 5,400 | 51,700 | DF & BB | ✗ | ✗ | ✗ |
| OmniBench | ARXIV'24 | I+A | ✗ | 1,142 | MC | ✗ | ✓ | ✓ |
| AV-Odyssey | ARXIV'24 | I+A | ✗ | 4,555 | MC | ✗ | ✓ | ✓ |
| WorldSense | ARXIV'25 | V+A | 1,662 | 3,172 | MC | ✗ | ✓ | ✓ |
| **Daily-Omni** | – | V+A | 684 | 1197 | MC | ✓ | ✓ | ✓ |

Our video data originates from AudioSet (Gemmeke et al., 2017), Video-MME (Fu et al., 2025a), and FineVideo (Farré et al., 2024). While AudioSet provides 10-second clips primarily for audio classification, we retrieved the original, full-length source videos. These were then processed into longer 30s or 60s segments which contains the original 10s segments to ensure they contain certain types of sound event. We employ Whisper-V3-Large (Radford et al., 2023) to ensure inclusion of spoken content. For Video-MME and FineVideo, our selection criteria prioritized videos containing substantial audio information alongside rich temporal visual dynamics. The inclusion of these datasets also increases diversity, as they feature more recent videos compared to Audio Set (primarily pre-2017 uploads), thus broadening the representation of genres and styles.

## 3.2 DATA ANNOTATION & QA CONSTRUCTION

We developed a pipeline that employs MLLMs to generate and revise visual and audio annotations. Concurrently, Reasoning Large Language Models (LLMs) are utilized to construct and optimize the associated questions, choices, and answers. To obtain detailed annotations while ensuring cost-effectiveness, we specifically used Gemini 2.0 Flash (Pichai et al., 2024) for the annotation task and Deepseek-R1 (DeepSeek-AI et al., 2025a) for QA construction and optimization. Figure 3 provides a outline of this process.

**Segment Annotation** Recognizing that even state-of-the-art Multimodal Large Language Models (MLLMs) can exhibit cross-modal hallucinations (Sung-Bin et al., 2025), we employed Gemini 2.0 Flash to annotate the audio and visual modalities independently. Additionally, since MLLM performance is known to degrade when processing long audio clips, we segmented the videos prior to annotation. Specifically, each clip was divided into three equal, shorter segments (e.g., 10s segments for 30s clips, 20s segments for 60s clips) to improve the MLLM's subsequent audio annotation quality.

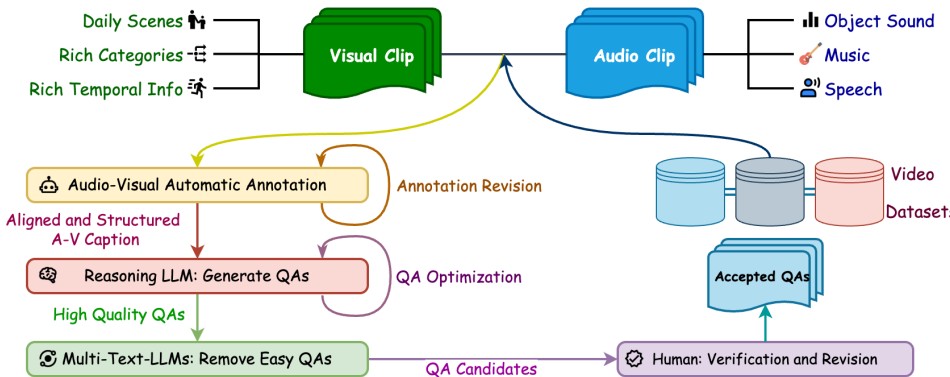

Figure 3: **The outline of Daily-Omni QA construction pipeline.** The arrows indicates the sequence of the processes.

**Visual & Audio Revision** After obtaining visual annotations from the video segments (processed without audio) and audio annotations from the corresponding audio segments, we use Gemini 2.0 Flash to perform a consistency check on the visual annotations. For this step, we prompt the model with the complete video clip, allowing it to review the segment annotations and ensure overall coherence. For example, referencing the full-length video, the model verifies whether a person described in an early segment is the same individual appearing in subsequent segments and generates a consistent revised annotation. Subsequently, audio annotations undergo refinement using a Reasoning LLM (Gemini 2.0 was employed in the initial phase of the project, with Deepseek-R1 used in later stages). This model leverages the consistent visual annotations to perform cross-modal correction, rectifying sound misidentifications and identifying sound sources. For example, if the audio is annotated as a 'generic impact sound' but the visual annotation for the same segment shows a 'door slamming shut', the Reasoning LLM uses this visual context to correct the audio description to 'door slamming sound' and attributes the sound's origin to the observed door.

**Visual & Audio Event Alignment** At this stage, we have sequences of visual and audio events annotated within consecutive 10s or 20s segments. However, these segment-level annotations do not explicitly specify the temporal alignment between individual visual and audio events – i.e., which specific events occurred simultaneously. To establish this precise cross-modal event concurrency, we proposed a novel **event aligning technique**. By prompting Gemini 2.0 Flash with the complete audio-visual clip, we instruct it to identify the visual event(s) occurring concurrently with each identified audio event. These aligned audio-visual event pairs provide sufficient information to infer the temporal relationships between any audio and visual event within the sequence. The details of annotation generation, revision and event alignment is shown in Figure 4.

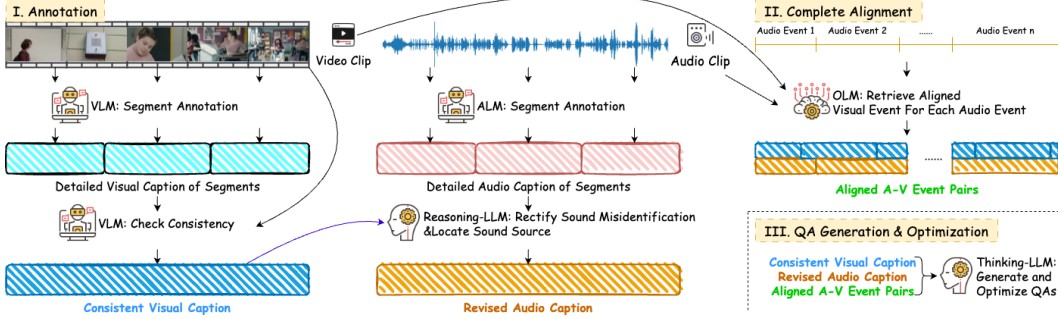

Figure 4: **Details of Daily-Omni annotation generation, revision and event alignment.** For cost-efficiency, we align all events with one query.

**QA Construction** Using the consistent visual annotations, revised audio annotations, and aligned event pairs derived from each video, we prompted Deepseek-R1 to generate multi-choice questions covering the following types: (1) AV Event Alignment: Questions to determine which audio and visual events occurred simultaneously with each other; (2) Event Sequence: Questions to determine the temporal sequence of visual and audio events in the video; (3) Reasoning: Questions to explain the

cause or reason behind the occurrence of a visual or audio event in the video; (4) Inference: Questions to speculate on information not explicitly presented in the video; (5) Comparative: Questions to compare the similarity or difference between the audio and visual information of two or more events in the video; (6) Context Understanding: Questions to determine the contextual information surrounding a specific event in the video. Since Daily-Omni aims to evaluate MLLM perception and reasoning within real-life audio-visual scenarios, we deliberately **exclude certain question types**. Specifically, counting and measuring questions are omitted, as they often rely on a single modality rather than integrated multi-modal understanding. Similarly, purely knowledge-based queries, such as celebrity identification, fall outside our intended scope.

**QA Optimization and Quality Control** By avoiding strict question templates, the generated QAs exhibit greater creativity, incorporate complicated logic, and sometimes feature obscure descriptions, making them more closely resemble questions asked in real-life scenarios. However, the generated questions and choices can sometimes contain excessive textual information, potentially allowing powerful models to infer the correct answer using text alone, without engaging with the audio-visual content. Therefore, Deepseek-R1 was employed again to remove superfluous textual information from the questions and choices. Additionally, it replaced obviously incorrect options with more challenging distractors, thereby increasing the difficulty and reducing the potential for correct answers based on guessing alone. Subsequently, we evaluated the optimized questions using two powerful LLMs, GPT-4o (OpenAI et al., 2024) and Deepseek-V3 (DeepSeek-AI et al., 2025b), providing them with only the textual questions and choices (no audio-visual context). Questions that could be answered correctly by both LLMs under this text-only condition were discarded, as they did not necessitate multimodal reasoning. This automated filtering step resulted in approximately 47% of the candidate QAs being discarded. Finally, the remaining QAs underwent manual evaluation for quality control. Human evaluators examined each QA, verifying: (1) that there was exactly one unambiguously correct answer among the choices, (2) that the proposed answer was indeed the correct one, and (3) that answering the question genuinely required comprehensive audio-visual capabilities. Based on this assessment, evaluators either accepted the QA for inclusion in the final benchmark or rejected it. Facilitated by the automated pipeline, the final human evaluation process was highly efficient. A **single annotator** used less than **30 hours** to review the candidates and establish the final set of 1197 QAs, corresponding to an acceptance rate of approximately **30%** during this manual review stage.

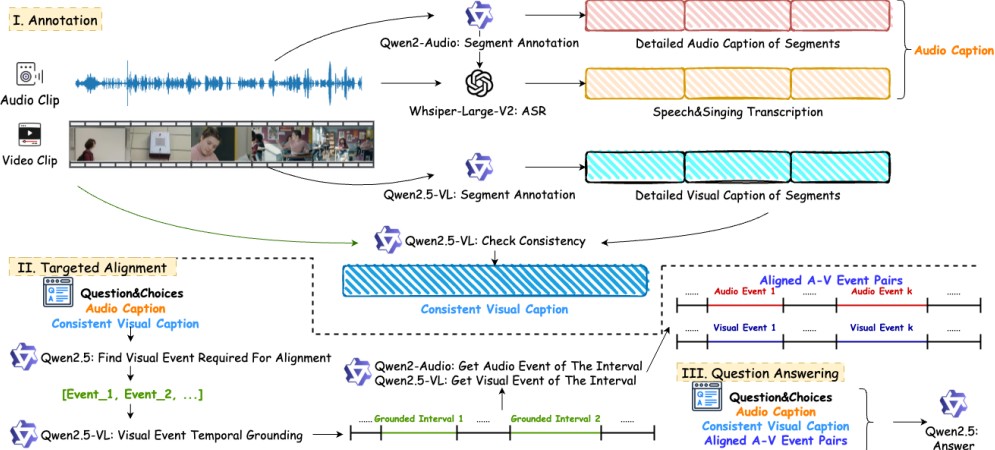

Figure 5: **The outline of Daily-Omni Agent workflow.**

### 3.3 DAILY-OMNI AGENT

To establish a baseline for MLLMs and demonstrate the importance of temporal awareness in audio-visual question answering, we constructed the **"Daily-Omni Agent"**. This agent, designed to understand audio-visual context and answer related questions, integrates several models: Qwen2-Audio (7B parameters) (Chu et al., 2024), Qwen2.5-VL-7B (Bai et al., 2025), Whisper-Large-V2 (Radford et al., 2023), and the text-based Qwen2.5-14B-Instruct (Qwen et al., 2025). As shown in Figure 5, when presented with a question, its choices, and the associated video context, the agent first divides both the video and audio streams into three segments of equal duration. Subsequently,

it independently generates annotations for these segments: visual annotations using Qwen2.5-VL and audio annotations using Qwen2-Audio. Additionally, we use Whisper-Large-V2 to provide a transciption of speech and singing of each segment, as Qwen2-Audio tends to omit this information in its annotations. Following the segment annotation, the agent utilizes Qwen2.5-VL to perform a **consistency check** on the visual annotations using the complete video, mirroring the revision process described previously (Section 3.2).

While generating aligned audio-visual event pairs would ideally provide richer temporal information for reasoning, implementing this step within the agent presents practical challenges. Unlike the data curation pipeline, the agent lacks access to a highly capable OLM like Gemini 2.0 Flash for precise event alignment. Furthermore, providing a large number of fine-grained event pairs risks overwhelming the context capacity or reasoning capabilities of the agent's LLM. Therefore, as an alternative to full event alignment, we adopt a innovative **targeted approach**: First, we prompt Qwen2.5-14B-Instruct with the visual and audio annotations, the question, and its choices. The model's task is to identify a list of specific **visual events** whose temporal localization is deemed necessary to answer the question correctly. Subsequently, we utilize Qwen2.5-VL-7B, functioning as a video temporal grounding model, to determine the start and end timestamps for each of these identified critical events. If the duration of a grounded event interval falls below a predefined threshold, the agent classifies this interval as critical. It then retrieves descriptions of both visual and audio events occurring within this specific, brief period using the previously mentioned approach, thereby establishing a localized alignment between concurrent events. Finally, we prompted Qwen2.5-14B-Instruct to determine the correct answer by providing it with the question, choices, the previously generated visual and audio annotations, and the extracted aligned event pairs.

## 4 EXPERIMENT

In this section, we present a comprehensive evaluation of recent Multimodal Large Language Models (MLLMs) on our benchmark to delineate their capability boundaries. Furthermore, we conduct ablation studies to investigate the key factors influencing model performance.

### 4.1 SETTINGS

Our evaluation encompassed three distinct types of models. Firstly, we examine **OLMs**, including several open-source contenders like VideoLLaMA 2 (Cheng et al., 2024), Unified-IO 2 (Lu et al., 2024b), Qwen2.5-Omni (Xu et al., 2025), Ola (Liu et al., 2025c) and our **Daily-Omni Agent**, as well as the proprietary Gemini models (Pichai et al., 2024). Secondly, we assess **VLMs** including Qwen2.5-VL (Bai et al., 2025) and GPT-4o (OpenAI et al., 2024) and **ALMs** including Audio Flamingo 3 (Goel et al., 2025) and Qwen2-Audio (Chu et al., 2024). We also evaluate the performance of some OLMs when provided with only visual inputs. Finally, we test some **LLMs** such as Deepseek-V3 (DeepSeek-AI et al., 2025b), GPT-4o (OpenAI et al., 2024) and Qwen2.5-14B(Qwen et al., 2025) without visual and audio inputs to check whether our benchmark contains too much information in questions and choices. All model evaluations are conducted strictly according to instructions from their respective official repositories and accompanying documentation (such as "cookbooks" or developer guides).

### 4.2 MAIN RESULTS

Table 2 presents comprehensive evaluation results, shedding light on the real-world audio-visual understanding capabilities of contemporary MLLMs and our proposed **Daily-Omni Agent**. Firstly, earlier Omni-modal Language Models (OLMs) such as Unified-IO 2 and VideoLLaMA 2 demonstrate limited performance on our benchmarks. Notably, their results are, in several instances, even surpassed by text-only LLMs. Furthermore, the Unified-IO 2 series displays a perplexing degradation in performance with increasing model size, an observation consistent with findings from OmniBench (Li et al., 2024). A plausible explanation is an inadequate capacity within these language models to effectively process and synthesize cross-modal information.

Secondly, while overall performance still presents challenges, more recent open-source OLMs such as the Qwen2.5-Omni series and Ola demonstrate reasonable proficiency in inference and reasoning tasks. This suggests a foundational understanding of general visual and audio contexts. However,

their efficacy significantly diminishes on temporally-sensitive tasks like audio-visual event alignment and context understanding, where results remain largely unsatisfactory. Despite incorporating cross-modal positional embeddings designed to provide temporal awareness, these models still struggle to address questions requiring precise temporal capabilities. Conversely, proprietary OLMs such as Gemini 2.0 Flash exhibit markedly superior cross-modal temporal capabilities, attaining the leading overall performance score of **67.84%**. The observation that even top-tier OLMs have not yet breached the 70% accuracy threshold underscores the demanding nature of this benchmark, positing it as a valuable and pertinent objective for the advancement of contemporary OLMs.

Table 2: **Performance comparison of MLLMs on Daily-Omni**. Boldface and underline indicate the top two performers. Subscripts on 'Avg' for visual-only OLMs show the performance drop from their audio-visual counterparts. Random guess accuracy is 25%. See Appendix B for evaluation details.

| Methods | AV Event Alignment | Comparative | Context Understanding | Event Sequence | Inference | Reasoning | 30s Subset | 60s Subset | Avg |
|---|---|---|---|---|---|---|---|---|---|
| **Omni-Modal Language Models (With Visual and Audio)** | | | | | | | | | |
| Qwen2.5-Omni (7B) | 44.12 | 51.15 | 38.86 | 40.52 | 57.79 | 61.71 | 46.68 | 48.36 | 47.45 |
| Qwen2.5-Omni (3B) | 38.66 | 48.09 | 33.68 | 33.99 | 54.55 | 44.00 | 42.35 | 38.36 | 40.52 |
| Ola (7B) | 40.34 | 61.07 | 40.41 | 43.46 | 63.64 | 69.71 | 51.47 | 49.82 | 50.71 |
| Unified-IO-2 L (1B) | 27.31 | 22.90 | 26.42 | 27.78 | 29.87 | 29.14 | 27.67 | 27.09 | 27.40 |
| Unified-IO-2 XL (3B) | 30.25 | 30.53 | 25.39 | 29.08 | 33.12 | 21.71 | 28.13 | 28.55 | 28.32 |
| Unified-IO-2 XXL (8B) | 25.63 | 31.30 | 26.42 | 25.82 | 35.06 | 29.71 | 26.74 | 30.00 | 28.24 |
| VideoLLaMA2 (7B) | 35.71 | 35.88 | 35.75 | 31.70 | 40.91 | 34.29 | 38.02 | 31.82 | 35.17 |
| Gemini 2.0 Flash | **62.18** | **73.28** | **63.73** | **63.72** | 76.62 | **75.43** | **67.23** | **68.55** | **67.84** |
| Gemini 2.0 Flash Lite | 55.04 | 64.89 | 58.03 | 54.25 | 74.03 | 72.00 | 62.44 | 60.00 | 61.32 |
| **Daily-Omni (ours)** | 51.68 | 68.70 | 60.10 | 53.92 | **78.57** | 71.43 | 63.99 | 59.27 | 61.82 |
| **Omni-Modal Language Models (Visual Only)** | | | | | | | | | |
| Qwen2.5-Omni (7B) | 38.24 | 48.85 | 34.72 | 36.27 | 51.95 | 45.71 | 40.80 | 41.64 | $41.19_{-6.3}$ |
| Qwen2.5-Omni (3B) | 33.61 | 42.75 | 36.27 | 33.33 | 49.35 | 38.86 | 38.79 | 36.55 | $37.76_{-2.8}$ |
| Gemini 2.0 Flash | 39.08 | 64.12 | 56.48 | 56.21 | 67.53 | 62.29 | 56.57 | 55.45 | $56.06_{-11.8}$ |
| Gemini 2.0 Flash Lite | 43.70 | 58.02 | 53.89 | 45.10 | 64.29 | 60.57 | 53.01 | 51.64 | $52.38_{-8.9}$ |
| **Visual Language Models (Visual Only)** | | | | | | | | | |
| GPT-4o | 47.90 | 62.60 | 52.33 | 52.61 | 66.23 | 66.29 | 55.64 | 57.45 | 56.47 |
| Qwen2.5-VL (7B) | 36.97 | 46.56 | 33.68 | 37.91 | 51.95 | 44.00 | 39.26 | 42.36 | 40.68 |
| Qwen2.5-VL (3B) | 35.71 | 43.51 | 34.72 | 33.66 | 43.51 | 39.43 | 37.71 | 37.09 | 37.43 |
| **Audio Language Models (Audio Only)** | | | | | | | | | |
| Audio Flamingo 3 (7B) | 40.76 | 55.73 | 43.01 | 40.52 | 65.58 | 68.00 | 50.23 | 49.45 | 49.87 |
| Qwen2-Audio (7B) | 28.99 | 35.88 | 27.46 | 32.03 | 33.77 | 33.14 | 31.22 | 31.82 | 31.50 |
| **Textual Language Models (Without Visual and Audio)** | | | | | | | | | |
| GPT-4o | 33.19 | 43.51 | 28.50 | 30.39 | 44.81 | 46.86 | 36.48 | 36.18 | 36.34 |
| Deepseek-V3 (671B) | 31.93 | 41.22 | 29.02 | 29.41 | 44.81 | 46.29 | 35.24 | 36.00 | 35.59 |
| Qwen2.5-Instruct (14B) | 30.25 | 39.69 | 27.98 | 28.43 | 42.21 | 42.86 | 32.15 | 35.82 | 33.83 |

Thirdly, upon ablating audio input and providing only visual data, all evaluated OLMs exhibit a substantial decline in performance. Notably, the magnitude of this performance loss tends to be greater for models that demonstrated superior initial performance when leveraging both modalities. The significant performance drop, especially in high-performing models, when deprived of audio, validates that the tasks within this benchmark genuinely require and benefit from the integration of both auditory and visual information, rather than being solvable by visual cues alone. This dependence on both modalities is further evidenced by the performance of leading Visual Language Models (VLMs). For instance, Qwen2.5-VL (7B) achieves only 40.68% accuracy, and even the highly capable GPT-4o (with its visual input) reaches just 56.47%. Given that these are state-of-the-art models within their respective size categories (sub-7B and larger proprietary models), their comparatively modest performance on our benchmark further underscores its value in demanding **genuine audio-visual integration**, a capability absent in previous multi-modal benchmarks.

These results reveal that: (1) Current OLMs demonstrate reasonable general audio and visual understanding capabilities when processing videos under 60 seconds in duration. However, they continue to struggle with tasks demanding sophisticated cross-modal temporal awareness or the integration of information across extended time intervals. (2) Our proposed question-answer generation pipeline proves effective in creating challenging evaluation instances that accurately probe the audio-visual understanding capabilities of MLLMs.

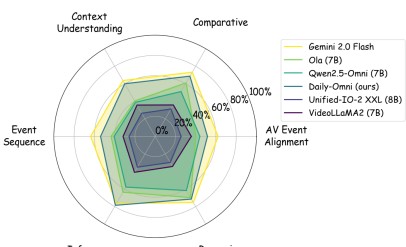

Figure 6: **MLLMs' accuracy over different question categories**

| Align Method | Average Accuracy | Aligned Event Pairs per Question |
|---|---|---|
| No Alignment | 60.65 | 0 |
| Naive Alignment | 59.65 | 0.68 |
| Smart Alignment | **61.82** | **1.11** |

Table 3: **Impact of different aligning methods on average accuracy and aligned event pairs per question.**

### 4.3 RESULTS OF DAILY-OMNI AGENT

Our Daily-Omni Agent, leveraging Qwen2.5-VL-7B, Qwen2-Audio (7B), and Qwen2.5-Instruct-14B, achieves **61.82%** overall accuracy. This performance is **state-of-the-art** among open-source methods and surpasses smaller proprietary MLLMs. The success of our agent suggests that by simply leveraging VLM and ALM with time-period-level alignment coupled with event-level alignment for some key events, we are able to create a very strong OLM.

To further study the effect of event alignment, we conducted an ablation study evaluating the Daily-Omni Agent's performance under three alignment scenarios: (1) **No Alignment**: Generating comprehensive visual and audio captions as a base, but omitting any form of event alignment. (2) **Naive Alignment**: Generating comprehensive visual and audio captions, followed by a step where the VLM is prompted to extract question-relevant events and their temporal boundaries (begin/end times). The ALM is then utilized to produce audio captions specifically for those extracted segments and attaining aligned audio-visual event pairs in the process. (3) **Smart Alignment**: Generating comprehensive visual and audio captions and aligned events as stated in Section **3.3**. Note that for (2) and (3), we only consider generating an align event pair when the provided segment has a duration below a certain threshold to prevent confusion. The overall accuracy and the average number of aligned event pairs identified per question for each method are presented in Table 3. As expected, the Smart Alignment method achieves the highest average accuracy. This demonstrates that explicitly identifying and aligning relevant events significantly boosts the model's overall performance compared to simply generating global captions. Conversely, the Naive Alignment method exhibited a slight decline in accuracy relative to the No Alignment baseline. Careful analysis of the generated events and their temporal boundaries suggests that this outcome is likely attributable to the limitations of the Qwen2.5-VL-7B model. It appears that this model is not sufficiently powerful to reliably identify and retrieve the target event through a single query. Moreover, the temporal grounding process itself frequently yields imprecise results, leading to erroneous alignment. This issue with temporal grounding affects even the Smart Alignment method, occasionally producing incorrect or confusing aligned event pairs. Consequently, we hypothesize that equipping the agent with a more powerful open-vocabulary video temporal grounding model would unlock further significant improvements in its performance, mitigating the impact of imprecise temporal grounding.

## 5 CONCLUSION

This paper introduced Daily-Omni, a novel Audio-Visual Question Answering benchmark designed to evaluate MLLMs on temporally-aligned multimodal reasoning in daily life scenarios. We also proposed an efficient Daily-Omni QA Generation Pipeline and the training-free Daily-Omni Agent, a strong open-source baseline. Our evaluation revealed that while recent MLLMs show general audio-visual understanding, they significantly struggle with precise cross-modal temporal awareness The results underscore our pipeline's value and effectiveness in creating such challenging questions. The Daily-Omni Agent, with its targeted temporal alignment, achieved competitive results, and our ablation study confirmed that a **Smart Alignment** strategy improves performance, unlike naive approaches. In summary, Daily-Omni highlights the current MLLM frontiers in complex audio-visual temporal reasoning. Future efforts should prioritize developing more accurate and robust multimodal temporal grounding techniques for truly sophisticated audio-visual understanding.

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

## A  CASE STUDY

This section delves into selected case studies of model performance on Daily-Omni. Such an analysis of specific examples offers deeper insights into both the nature of the Daily-Omni benchmark and the current capabilities and limitations of the models. Figure 7 presents three illustrative questions from the Daily-Omni benchmark. It displays the responses of the Gemini 2.0 Flash and Qwen2.5-Omni models under two conditions: processing full audio-visual inputs versus visual-only inputs.

**Case 1: Chronological Event Ordering** This question requires determining the first event to occur in the video from a given list of audio-visual events. To correctly ascertain the existence and chronological position of choices B (Player discussing a personal milestone) and D (Female presenter

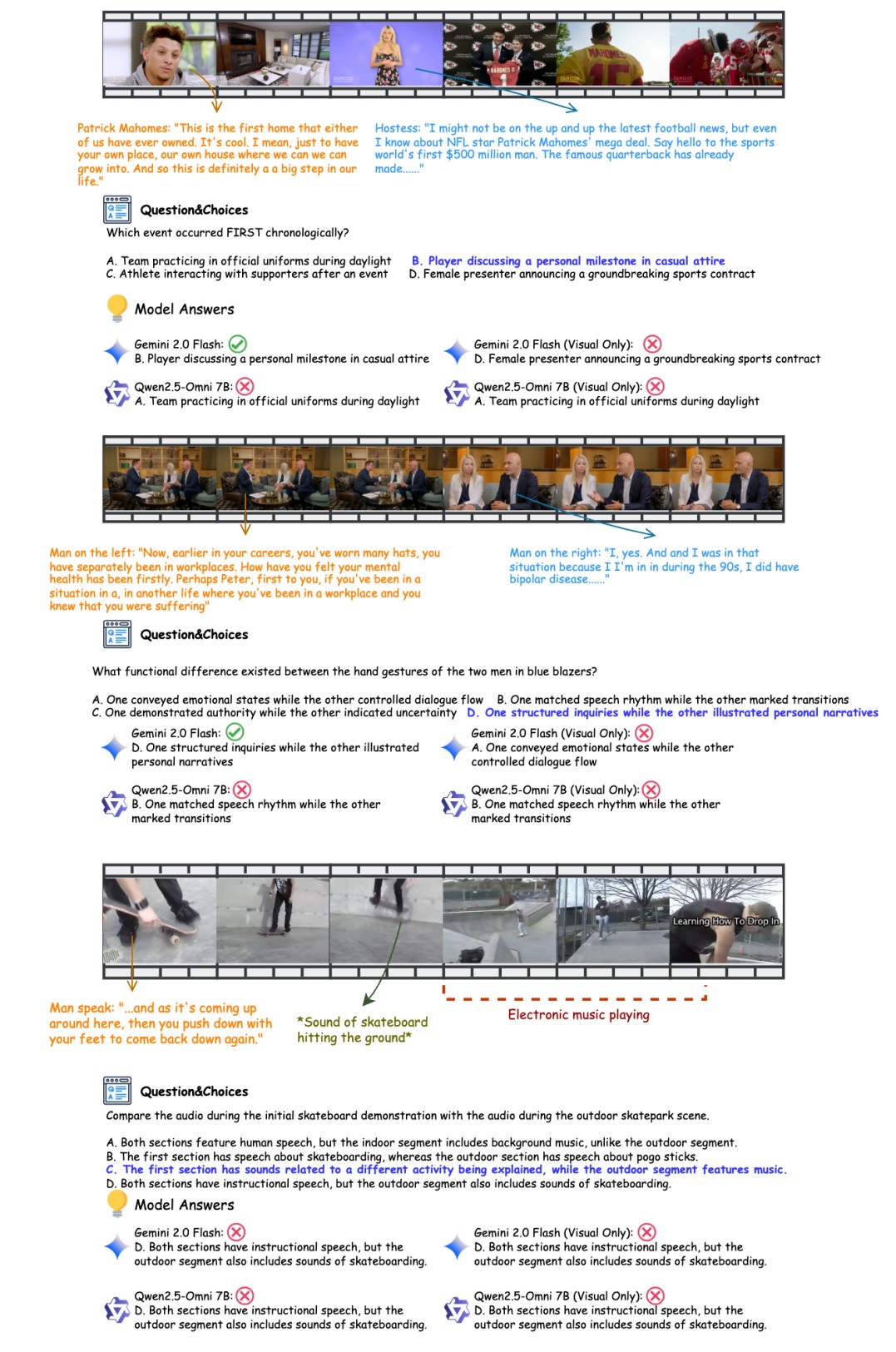

Figure 7: **OLM responses to Daily-Omni questions: A comparison of audio-visual versus visual-only inputs.** The figure presents examples of OLM performance when using full audio-visual modalities compared to visual-only input. For each case, it shows the audio-visual content, the question with choices (correct answer highlighted), and the models' answers, indicating correctness for each input condition.

announcing a groundbreaking sports contract), the model must leverage both audio and visual sensory inputs. Gemini 2.0 Flash successfully identifies choice B as the initial event, whereas Qwen2.5-Omni 7B fails to do so. Both model failed to answer correctly when only taking visual input. This disparity suggests that merely possessing multi-modal sensory capabilities is insufficient; models also require robust cross-modal reasoning to effectively integrate audio-visual information and achieve correct temporal ordering.

**Case 2: Audio-Visual Inference** This case assesses the ability to discern the functional differences between the hand gestures of two individuals engaged in a discussion. While visual input captures the gestures themselves, audio input is also critical for interpreting their communicative purpose as it reveals what is being said and who said it. Gemini 2.0 Flash correctly identified the functional difference (D) with audio-visual input but failed with visual-only input, selecting a more generic but incorrect interpretation (A). Qwen2.5-Omni 7B, failed to make the correct distinction in both audio-visual and visual-only conditions, selecting an unreasonable option (B). This question underscores that merely understanding the speech content is insufficient; accurately identifying the speaker for each utterance is also crucial. Successfully attributing speech requires robust audio-visual temporal integration—the ability to precisely synchronize the spoken audio with the visual depiction of the active speaker. Following this, strong reasoning ability is needed to infer the distinct roles and communicative intents (e.g., inquirer vs. narrator) based on the combined audio-visual information, and thereby understand the different functions of their gestures. The failure of Qwen2.5-Omni 7B, even with audio-visual input, suggests potential limitations in one or both of these sophisticated cross-modal capabilities.

**Case 3: Sound Understanding** This question directly tests the model's ability to analyze and compare audio characteristics across two different segments of a video-an skateboard demonstration and an outdoor skatepark scene. The task involves identifying distinct audio elements such as human speech, specific sound effects (e.g., skateboard impact), and background music. This is fundamentally an audio-centric task where visual context primarily helps situate the sounds. Strikingly, both Gemini 2.0 Flash and Qwen2.5-Omni failed to correctly compare the audio content, selecting the same incorrect option (D) irrespective of whether they received audio-visual or visual-only input. Their failure in the audio-visual condition suggests a significant challenge in fine-grained auditory scene analysis, such as distinguishing non-verbal sound event and music, or accurately mapping these perceived audio features to the descriptive choices provided.

# B    EVALUATION DETAILS

The models listed in Table 2 were evaluated as follows: Qwen2.5-Omni (7B and 3B), Ola, Unified-IO-2 (L, XL, and XXL), VideoLLaMA2, and Qwen2.5-VL (7B and 3B) were deployed and tested on a local server. In contrast, Gemini 2.0 Flash, Gemini 2.0 Flash Lite, GPT-4o, and Qwen2.5-Instruct (14B) were evaluated via their respective APIs.

For the Daily-Omni Agent, Qwen2.5-VL-7B-Instruct and Qwen2.5-14B-Instruct were accessed via the API provided by Alibaba Cloud's Bailian platform (accessible at https://bailian.console.aliyun.com/) to ensure operational efficiency. Similarly, Whisper-Large-V2 was utilized via the API provided by OpenAI. The Qwen2-Audio component, on the other hand, was deployed and tested locally. It is important to note that all component models of Daily-Omni are capable of local deployment. According to official documentation from Aliyun (https://help.aliyun.com/zh/model-studio/model-user-guide/) and OpenAI (https://openai.com/index/introducing-chatgpt-and-whisper-apis/), the Qwen models offered through Bailian and the Whisper-Large-V2 model accessed via the OpenAI API are identical to their respective open-source or standard versions. Therefore, no performance differences are anticipated.

Our code implements direct passing of local video path to the Qwen2.5-VL API. However, this functionality might require you to contact Aliyun customer service to enable. If direct path input is not activated, you can alternatively pass a list of video frames, though this may result in suboptimal performance.

