# OpenReview forum: "Daily-Omni: Towards Audio-Visual Reasoning with Temporal Alignment across Modalities"
_ICLR.cc/2026/Conference — Submitted to ICLR 2026_

### Official Review · Reviewer_tGax · 2025-10-19

**Soundness:** 2
**Presentation:** 3
**Contribution:** 2
**Rating:** 4
**Confidence:** 5

**Summary:**

This paper introduces Daily-Omni, a new audio-visual question answering (AVQA) benchmark comprising 684 real-world daily-life videos and 1,197 multiple-choice QA pairs spanning six diverse tasks, from audio-visual event alignment to complex cross-modal reasoning. The authors also propose a scalable Daily-Omni QA Generation Pipeline with five automated modules, enabling efficient annotation with high quality (30% acceptance rate) and minimal human effort (~30 hours for one annotator). To establish a strong open-source baseline, they present the Daily-Omni Agent, which integrates off-the-shelf visual language models, audio language models, and automatic speech recognition systems without fine-tuning. Experiments show this agent achieves state-of-the-art performance among open-source methods, while also revealing that current multimodal large language models (MLLMs) still struggle with tasks requiring deep audio-visual temporal integration. The work highlights the effectiveness of modular, alignment-based approaches and provides a valuable resource for advancing multimodal reasoning research.

**Strengths:**

- This paper is well written and easy to follow.
- The authors devise a daily-omni QA generation pipeline to curate an audio-visual question answering dataset, providing a new AVQA dataset to the research community.

**Weaknesses:**

- The paper omits several relevant large models in the audio-visual question answering (AVQA) domain, such as VITA [1]. VITA is a multimodal large language model specifically designed for AVQA tasks. The authors could strengthen their empirical evaluation by including VITA in their benchmark comparison to better contextualize the performance of their proposed approach.

- The related work section overlooks several important AVQA datasets, including FortisAVQA [2] and MUSIC-AVQA-R [3]. These datasets enable more comprehensive and fine-grained evaluations. For instance, FortisAVQA [2] supports generative AVQA evaluation rather than being limited to multiple-choice questions. Given the rapid progress in large language models, generative answering capabilities, as demonstrated in works like [2] are increasingly aligned with real-world application requirements. In contrast, multiple-choice QA may not fully capture the reasoning and generation abilities of modern multimodal large language models (MLLMs).

- The novelty of the Daily-Omni Agent pipeline appears limited. It reads more like an engineering integration of existing components rather than a methodological contribution. The paper does not clearly articulate what conceptual or technical innovation this framework introduces beyond modular composition.

- There are a few minor typos. For example, in Section 1, the second line contains the phrase "of and interactions," which seems grammatically incorrect and should be revised.

References:
[1] https://github.com/VITA-MLLM/VITA
[2] FortisAVQA and MAVEN: A Benchmark Dataset and Debiasing Framework for Robust Multimodal Reasoning
[3] Look, Listen, and Answer: Overcoming Biases for Audio-Visual Question Answering

**Questions:**

- In Section 3.2, what is the sampling interval used to segment the videos? Specifically, for a 30-second clip, how are the 10 segments selected? Are they sampled at fixed intervals, or is another strategy (e.g., random or content-aware sampling) employed?

- Also in Section 3.2, the paper states that a single annotator can complete quality filtering for all 1,197 QA pairs within 30 hours. Can the authors clarify how this single-annotator process ensures consistent and reliable dataset quality? Was inter-annotator agreement measured or pilot testing conducted to validate annotation reliability?

- I observe that the Daily-Omni Agent workflow closely mirrors the pipeline used for dataset curation (e.g., leveraging ASR, VLM, and ALM with temporal alignment). Does this imply that the agent’s strong performance on the Daily-Omni benchmark is partly due to the alignment between the data generation process and the agent’s design? In other words, could the benchmark be inadvertently biased toward this specific modular architecture?

---

> ### Author Response · Authors · 2025-11-18
>
> Thanks for your detailed review. We would like to address the weaknesses and questions you raised point by point.
> ### Weakness 1: On VITA
> Thank you for highlighting the importance of VITA as a significant model in the AVQA domain. We would like to respectfully clarify that we were indeed aware of this work and have already cited VITA 1.5 at the end of **Section 2.1**.
> The primary reason VITA 1.5 was not included in our experimental benchmark is due to a fundamental mismatch between its supported input modalities and the requirements of our Daily-Omni tasks. As we noted in our paper, VITA 1.5 belongs to a category of models where the audio encoder is primarily designed to process user instructions (i.e., spoken questions), functioning as an alternative to the text modality for the query, rather than perceiving the environmental context.
> Specifically, the official VITA 1.5 model supports two main input configurations: video + audio instruction or video + text question. It is not designed to simultaneously process three distinct streams: a video stream, an accompanying environmental audio stream, and a textual question. Since the core challenge of Daily-Omni is to reason about the synchronized events between video and environmental audio, VITA's architecture is not suited for direct evaluation on our benchmark.
> To prevent any future confusion, we will expand our discussion in the final version of the paper to make this architectural distinction and the reason for VITA's exclusion from our experiments more explicit. We thank you for prompting this important clarification.
>
> ### Weakness 2: On Dataset Complementarity and the Rationale for MCQ
> Thank you for pointing out these important AVQA datasets. We agree that FortisAVQA and MUSIC-AVQA-R are significant contributions, and we will add a detailed discussion of them to our related work section. Their approach, particularly in using LLMs to systematically revise existing QA pairs to mitigate dataset biases, is an elegant and effective strategy for enhancing benchmark robustness.
>
> Our work similarly embraces the power of LLMs within our data generation pipeline, but with a different focus. Instead of revising existing questions, we generate them from scratch without a fixed template to ensure diversity. More importantly, our process includes two rigorous, LLM-driven validation stages (details in Section 3.2):
> **QA Optimization**: We use an LLM not just to generate, but to refine the questions by adding more plausible and challenging distractor options, demanding a deeper level of reasoning.
> **Quality Control**: We employ text-only LLMs as an automated filter to discard any question that can be answered correctly without the audio-visual context, directly preventing the kind of linguistic "shortcut" biases that FortisAVQA aims to address.
>
> Regarding your insightful point on the evaluation format, our choice of multiple-choice questions (MCQ) was a deliberate decision to ensure the benchmark's objectivity and focus. While open-ended generation is a crucial capability, its evaluation often relies on powerful external models like GPT-4o as judges. This can introduce its own set of biases and, critically, conflates the assessment of multimodal reasoning with the model's text generation abilities.
> Given that the primary aim of Daily-Omni is to specifically evaluate an MLLM's core multimodal reasoning capabilities, we opted for the objectivity of the multiple-choice format. **It provides a clear, direct, and reproducible measure of whether the model has correctly understood the audio-visual context, which is the central focus of our study.**

---

> ### Author Response · Authors · 2025-11-18
>
> ### Weakness 3: On the Agent's Contribution as a Diagnostic Tool
> We appreciate the reviewer's issue.
> The agent's key conceptual innovation lies in its explicit and intelligent handling of audio-visual temporal alignment. This is not merely an implementation detail but the central hypothesis we tested. Our ablation study in Section 4.3 provides clear evidence for this:
> **Smart Alignment** (aligning only question-related events) significantly outperforms the **No Alignment** baseline.
> Conversely, **Naive Alignment** (aligning all detected events) actually degrades performance, demonstrating that simply providing more multimodal information is not enough; the quality and relevance of the alignment are paramount.
> **This finding—that intelligent, selective cross-modal alignment is a key driver of performance—is a concrete methodological insight.**
> This conclusion is further reinforced by the overall benchmark results on other MLLMs. Models like Qwen2.5-omni, which incorporates TMRoPE for explicit cross-modal positional embedding, show superior performance compared to models like VideoLLaMA2 that lack such sophisticated alignment mechanisms. This shows that our benchmark successfully identifies alignment as a critical capability.
> Therefore, the contribution of the Daily-Omni Agent is twofold:
> 1) It establishes a strong, transparent performance baseline for multimodal reasoning using unimodal expert models.
> 2) More importantly, it serves as a diagnostic tool that reveals the critical importance of intelligent modal alignment, providing a clear and valuable directive for future omni-modal model design.
>
> ### Weakness 4: Regarding Minor Typos
> Thank you for pointing this out. We appreciate your careful reading. We will thoroughly proofread the entire manuscript and correct this specific error, along with any other grammatical mistakes, in the final version.
>
> ### Question 1: Clarification on Video Segmentation
> Thank you for this question.
> For the data annotation stage described in Section 3.2, we segment the videos into continuous, non-overlapping clips using fixed intervals. This contiguous approach was chosen because our goal is to capture **all relevant information** across the entire video duration, ensuring no potential audio-visual events are missed by a sampling strategy.
> To clarify the detail you asked about, a 30-second video is divided into three 10-second segments for annotation: 0-10s, 10-20s and 20-30s.
> Each of these segments is processed to ensure comprehensive event captioning. This same strategy is applied to the 60-second videos (i.e., three 20-second segments). We will revise Section 3.2 in the final version to make this process and its rationale more explicit. Thank you for helping us improve the paper's clarity.
> ### Question 2: On Annotation Reliability and Consistency
>
> Thank you for thisquestion regarding our annotation and quality control process.
> While inter-annotator agreement (IAA) is the standard for tasks requiring subjective interpretation by multiple people, our final stage was a filtering task based on a strict, pre-defined set of criteria. To ensure the reliability of this process, we took the following steps:
> Establishment of a Clear Rubric: Before filtering, our team collaboratively defined a rigorous set of acceptance criteria, as outlined in **Section 3.2**. A QA pair was accepted only if it met all three conditions: (1) it had exactly one unambiguously correct answer, (2) the provided answer was indeed correct, and (3) answering the question genuinely required comprehensive audio-visual reasoning.
> Single Annotator for Consistency: Once this high-fidelity standard was established and calibrated across the team, **the final filtering was performed by a single, trained author.** The primary reason for this choice was to maximize intra-annotator consistency (i.e., ensuring the same standard is applied uniformly across all 1,197 samples) and avoid any potential drift in standards between different people over a long annotation period.
> To make this process efficient and accurate, we developed a simple Gradio UI that displayed the video clip alongside the question and options. The annotator could then efficiently accept or reject the sample based on the established rubric.

---

> ### Author Response · Authors · 2025-11-18
>
> ### Question 3: On Architectural Alignment
>
>  Thanks for your insightful question. We acknowledge the structural similarity, but we argue this stems not from an inadvertent bias, but from the intrinsic nature of the task itself.
> At its core, any form of complex multimodal reasoning, whether performed by humans or AI, involves two fundamental stages:
> - Perception and Alignment: First, one must perceive the distinct audio and visual events and correctly align them in time.
> - Reasoning: Second, one must use this aligned, multimodal information to reason about relationships and answer a specific question.
> Our Daily-Omni Agent simply makes these universal steps explicit through its modular design. The similarity in structure between the agent and the data pipeline exists because they both follow this fundamental logic required to solve the task.
>
> Crucially, the potential for architectural bias is significantly mitigated because the final QA pairs in our benchmark are not a direct product of the initial annotation process. They undergo three rigorous stages of refinement that decouple them from the initial structured data:
> **QA Optimization** adds linguistic complexity and challenging distractors.
> Automated **Quality Control** filters out questions with "shortcut" biases.
> **Human Filtering** ensures the final questions are unambiguous and genuinely require deep reasoning.
>
> These steps significantly alter the final questions, ensuring they are not a simple reverse-engineering of the initial data generation logic. Therefore, the benchmark is not biased towards our agent's specific architecture, but rather towards the fundamental capability of performing accurate audio-visual reasoning. The agent's strong performance is a key finding: it demonstrates that an explicit, decomposable approach to this universal process is currently more effective than the implicit strategies learned by end-to-end models.
>
> **We hope our clarifications have addressed your concerns, and we would be grateful if you would reconsider your evaluation of our paper's contribution.**

---

### Official Review · Reviewer_YCPb · 2025-10-27

**Soundness:** 3
**Presentation:** 2
**Contribution:** 2
**Rating:** 2
**Confidence:** 4

**Summary:**

This paper explores the ability of Multimodal Large Language Models (MLLMs) to perform synchronized audio-visual reasoning. It introduces the Daily-Omni benchmark with 684 daily-life videos and 1,197 QA pairs across six tasks, the Daily-Omni QA Generation Pipeline for scalable and efficient QA creation, and the Daily-Omni Agent, a training-free model combining VLMs, ALMs, and ASR through temporal alignment. Experiments show that existing MLLMs struggle with audio-visual integration, while simple temporal alignment notably enhances multimodal reasoning performance.

**Strengths:**

1. Compared with WorldSense, this paper introduces a new benchmark dataset, an automated QA generation pipeline, and a training-free agent. Although the improvements are incremental, they hold practical significance within the field of multimodal research.
2. The proposed Daily-Omni benchmark covers diverse daily-life scenarios from multiple sources, including music, speech, and various environmental sound events. It also provides a complementary QA generation pipeline, offering a useful tool for future data creation and extension.
3. The paper evaluates multiple MLLMs on the Daily-Omni benchmark, demonstrating its challenging nature, while the proposed Daily-Omni Agent achieves state-of-the-art performance among open-source methods.

**Weaknesses:**

1. The paper presents a multi-stage data annotation and QA construction process based on Gemini 2.0 Flash and Deepseek-R1, which, while complete, functions more as a systematic workflow connected primarily through prompt engineering.

(1) Although each 30-second video is divided into three 10-second segments and each 60-second video into three 20-second segments, the entire process still relies heavily on Gemini 2.0 Flash’s interpretation of these clips. The method for aligning visual and audio events simply involves feeding the full audio-visual segment into Gemini 2.0 Flash and asking the model to identify corresponding visual events for each audio cue, essentially constituting a straightforward application of existing models.

(2) The QA construction stage relies on simple prompt-based generation using Deepseek-R1, which appears overly direct and lacks methodological depth.

(3) While the resulting Daily-Omni QA is presented as an automated data generation pipeline, the final human filtering process retains only about 30% of the generated samples, indicating that the automatically produced content remains inconsistent in quality and still requires substantial human effort.

2. The implementation of the Daily-Omni Agent primarily builds on existing model capabilities in a relatively straightforward and procedural manner. After dividing the video and audio streams into three equal temporal segments, the agent applies pre-trained models for event alignment and audio-visual understanding. While the authors note that WorldSense lacks explicit training strategies or architectural guidance for improving model performance, the proposed approach similarly does not provide concrete solutions to this issue.

3. I think that the state-of-the-art performance of the Daily-Omni Agent largely stems from its extensive use of multiple tools and models, such as Qwen2.5-VL and Qwen2.5-14B-Instruct, through repeated and combined applications. In contrast, other models in the comparison are evaluated directly on the benchmark without similar multi-tool integration or repeated inference.

**Questions:**

Is it sufficient to systematically evaluate a model’s performance across different temporal scales using only 30-second and 60-second clips? In fact, the difference between 30 and 60 seconds remains within the same order of magnitude, which limits the benchmark’s ability to assess model behavior over truly varying temporal durations ranging from seconds to hours.

---

> ### Author Response · Authors · 2025-11-18
>
> Thanks for your detailed review. We would like to address the weaknesses and questions you raised point by point.
> ### Weakness 1: On the Data Generation Pipeline
> #### (1) On AV Event Alignment:
> We agree with the reviewer that leveraging Multimodal Large Language Models (MLLMs) to generate video/audio captions is an established technique. However, we would like to clarify that our task—audio-visual event alignment—is more complex than generating a single, holistic caption. Before adopting our current approach, we thoroughly investigated several alternatives.
> - Open-Vocabulary Grounding Models: One potential method was to use open-vocabulary video and audio grounding models to identify the precise timestamps of events in each modality and then align them. However, the lack of accurate open-vocabulary audio grounding models made this a significant bottleneck, making the approach infeasible for scalable data creation.
> - Embedding-Based Alignment: We also considered aligning video frames and audio segments via their joint embedding space using models like ImageBind, CLIP, or CLAP. This approach, while theoretically sound, presented two major drawbacks in practice: prohibitively high inference costs due to the need for dense, frame-by-frame comparisons, and suboptimal performance in capturing the nuanced, event-level synchrony required for our benchmark.
>
> In contrast to these alternatives, our empirical evaluations showed that using Gemini 2.0 Flash offered the optimal trade-off between annotation quality and inference efficiency. Crucially, this step in our pipeline achieves a high accuracy of over **90% in correctly identifying and aligning audio-visual events, as verified by our manual checks.**
> Therefore, our contribution is not the mere application of an MLLM, but the design and validation of a system that efficiently produces structured annotations, including distinct audio captions, corresponding visual captions, and their temporal alignment. **We argue that building a tool that reliably accomplishes this complex task is, in itself, a meaningful contribution to the research community.**
>
> #### (2) On QA Construction:
> We would like to respectfully clarify that our method extends significantly beyond a simple, one-shot generation step with Deepseek-R1. As detailed in Section 3.2 of our paper, the initial generation is just the first part of a deliberate, multi-stage pipeline designed to ensure the quality, complexity, and fairness of the final question-answer pairs.
> Specifically, we implemented two crucial subsequent stages:
> - **QA Optimization**: Following the initial draft, we employ a QA Optimization step. This stage leverages Deepseek-R1 again, but in a different capacity—not as a generator, but as a refiner. It is specifically prompted to enhance the linguistic complexity of the questions, reduce potential ambiguities, and, most importantly, create more plausible and challenging distractor options for the multiple-choice format. This ensures that the questions are not trivial and require a deeper level of understanding.
> - **Automated Quality Control**: To rigorously validate the questions, we introduced a crucial Quality Control filter. This stage uses multiple LLMs to simulate a test-taker that has no access to the audio-visual context. We task these models to answer the questions based solely on the text. If the models can consistently guess the correct answer through common sense or linguistic biases alone, the question is flagged as a "shortcut" and discarded.
>
> Therefore, our QA construction process, far from being "overly direct," incorporates dedicated stages for both refinement and validation.
> #### (3) On the 30% Retention Rate:
> We would like to clarify that the 30% retention rate is not an indicator of a flawed pipeline, but a direct result of our strict quality standards for creating a high-quality benchmark.
> To maximize efficiency, our QA generation stage prompts Deepseek-R1 to generate multiple questions in a single pass. It is an expected outcome that the quality of these batched outputs will vary. This initial variance is precisely why our human filtering stage is so crucial and rigorous. We discard not only incorrect samples but also those that are ambiguous, overly simple, or too formulaic.
> This stringent curation ensures that the final Daily-Omni benchmark is a reliable and genuinely challenging tool for the community.

---

> ### Author Response · Authors · 2025-11-18
>
> ### Weakness 2: On the Daily-Omni Agent's Simplicity
> We appreciate the reviewer's characterization of our agent as a procedural framework. We would like to clarify that this design was intentional. The primary purpose of the Daily-Omni Agent is to serve as a strong, training-free baseline built from powerful unimodal components, allowing us to diagnose key challenges in multimodal reasoning without the confounding variable of end-to-end training.
>
> While our approach may not be a new model architecture, it absolutely provides concrete guidance on how to improve future models. This is demonstrated in our ablation study in Section 4.3, where we compare different alignment strategies for the agent:
> "Smart Alignment," which selectively aligns only question-relevant events, significantly improves performance over the "No Alignment" baseline.
> Crucially, "Naive Alignment," which aligns all detected events, actually degrades performance.
> **This comparison offers a clear and concrete insight for the field: effective omni-modal reasoning requires intelligent, selective cross-modal alignment, not just the co-location of multimodal information in the context.**
>
> This conclusion is further reinforced by the overall benchmark results. Models like Qwen2.5-Omni, which utilizes TMRoPE for explicit cross-modal positional embedding, outperform models that lack such sophisticated temporal alignment mechanisms (e.g., VideoLLaMA2).
> Therefore, while our agent is training-free, its design and the corresponding analysis serve a critical purpose by pinpointing precise A-V temporal alignment as a key bottleneck and a crucial area for future architectural innovation.
>
> ### Weakness 3: On the Fairness of the Comparison
> We appreciate the reviewer's concern.
> As mentioned in our previous response, a primary purpose of the Daily-Omni Agent is to establish a powerful performance baseline for future omni-modal LLMs, demonstrating what is achievable with existing, specialized components.
> While our agent indeed uses more modules and a multi-step process, we see this not as an unfair advantage, but as a key finding of our work. Its state-of-the-art performance proves the significant potential of multi-tool integration and repeated inference for solving complex multimodal reasoning tasks. This result highlights that current end-to-end models lack these explicit, multi-step reasoning capabilities, and it provides a clear direction for improving the next generation of omni-modal architectures.
>
> ### Question on Video Duration:
> This is an excellent and insightful question regarding the temporal scope of our benchmark. Our decision to focus on the 30-second and 60-second range was a pragmatic one, guided by two primary constraints in the current landscape of multimodal research:
>
> **Technical and Hardware Limitations**: As of now, many state-of-the-art MLLMs are architecturally or computationally limited. They either do not natively support video inputs spanning several minutes or hours, or processing such long videos would require prohibitive VRAM resources. This makes a systematic evaluation on longer-form videos infeasible for most researchers and impractical for establishing a widely-usable benchmark today.
>
> **The Challenge of Meaningful QA Creation**: There is a significant challenge in constructing meaningful question-answer pairs for very long videos (e.g., several minutes or hours). We observed that for most long-form content, the critical information required to answer a specific, targeted question is often localized within a brief segment of tens of seconds. Crafting questions that genuinely require reasoning over multiple, disparate events spread across a very long timeline is a non-trivial data creation problem in itself.
>
> Therefore, we positioned Daily-Omni as a foundational benchmark to address the immediate, observable challenge where even top models struggle when scaling from 30 to 60 seconds. Before the community can effectively tackle hour-long reasoning, it is essential to first solve the problems of temporal synchronization and information integration within this fundamental, sub-minute timescale. We agree that extending this research to longer durations is a critical next step, and we will highlight this as a key direction for future work.
>
>
> **We hope our clarifications have addressed your concerns, and we would be grateful if you would reconsider your evaluation of our paper's contribution.**

---

> ### Comment · Reviewer_YCPb · 2025-11-27
>
> Thank you to the authors for clarifying several of my concerns. Although the Daily-Omni QA Generation Pipeline and the Daily-Omni Agent are fundamentally workflow integrations of existing high-performance tools, and despite lacking methodological guidance on enhancing model capabilities via explicit training protocols or architectural modifications (a shortcoming the authors themselves noted in WorldSense), the work represents a relatively complete engineering implementation within the DB track. I will consider an appropriate increase in my score.

---

### Official Review · Reviewer_1BVk · 2025-10-30

**Soundness:** 3
**Presentation:** 3
**Contribution:** 3
**Rating:** 6
**Confidence:** 4

**Summary:**

This paper proposed Daily-Omni, a new AVQA benchmark designed to evaluate the synchronous, cross-modal reasoning capabilities of MLLMs. The proposed scalable, MLLM-assisted pipeline allows for a substantial reduction in data construction costs. Additionally, the authors propose Daily-Omni-Agent, an agent-based MLLM that achieves superior performance over open-source MLLMs on the proposed benchmark. Further comparative analysis reflects that existing audio-visual models still have significant room for improvement in audio-visual cross-modal understanding. By comparing the performance of the omni-model, visual-only models, and text models, it is demonstrated that Daily-Omni indeed requires joint audio-visual reasoning capabilities.

**Strengths:**

- The paper is well-written and easy to understand, the designed pipeline is reasonably structured, and the performance achieved by the proposed Daily-Omni-Agent reflects the current limitations of existing MLLMs.
- The comparative analysis across different models validates the importance of audio for video comprehension, thereby underscoring the necessity of developing an appropriate benchmark.

**Weaknesses:**

- I noticed that an earlier work, AVUT [1], s highly relevant to the research presented in this paper. The authors should provide a comparative analysis in the manuscript to clearly distinguish the contributions of this work from the prior study.
- This benchmark's reliance on data drawn solely from existing datasets is problematic for two reasons. 1) It potentially undermines the validity of the test, as the data may have been seen or is out of its original context. 2) It bypasses the critical, foundational challenge of raw data sourcing and filtering. The quality of a benchmark is heavily dependent on its raw data, and an ideal data construction pipeline should feature an automatic or semi-automatic mechanism for this curation.

[1] Yang et al., "Audio-centric Video Understanding Benchmark without Text Shortcut", EMNLP 2025.

**Questions:**

I am curious whether the authors plan to extend the evaluation to an open-ended format. If so, do they have any insights regarding the benchmark design and scoring mechanism for such a setup? (This question does not affect my assessment of the paper.)

---

> ### Author Response · Authors · 2025-11-18
>
> Thanks for your detailed review. We would like to address the weaknesses and questions you raised point by point.
>
> ### Weakness 1: On Differentiating from AVUT and Highlighting Our Contributions
> Thank you for bringing the highly relevant work, AVUT, to our attention. We agree that it is an important study in this domain, and we will add a thorough comparative analysis to our related work section in the final manuscript.
> While both our work and AVUT aim to create benchmarks that require genuine multimodal understanding, Daily-Omni introduces several distinct contributions:
>
> **Explicit Audio-Visual Event Alignment**: A core novelty of our work is the dedicated audio-visual event alignment step in our annotation pipeline. This allows us to specifically generate questions that probe a model's ability to reason about the fine-grained temporal synchronization and causal relationships between what is seen and heard—a dimension that is the central focus of our benchmark.
>
> **Advanced LLM-driven QA Generation and Refinement**: Our pipeline leverages the advanced capabilities of modern LLMs not only for initial question generation but also for subsequent QA Optimization (e.g., creating challenging distractors) and Quality Control (filtering out text-based shortcuts). This ensures a high level of reasoning complexity and diversity in the final benchmark.
>
> **Introduction of the Daily-Omni Agent**: Beyond the benchmark itself, we propose the Daily-Omni Agent, a strong baseline constructed from powerful unimodal components. This agent serves as a crucial tool for the community, demonstrating a highly effective, training-free approach for this task and providing a clear performance target for future end-to-end MLLMs.
>
> We see Daily-Omni as a valuable complement to AVUT, pushing the evaluation of audio-visual understanding towards more complex, temporally-aligned reasoning. We appreciate you prompting this comparison and will ensure it is clearly articulated in the revised manuscript.

---

> ### Author Response · Authors · 2025-11-18
>
> ### Weakness 2: On the Rationale and Validity of Using Existing Datasets
> Thank you for raising these important and nuanced points about benchmark creation. We appreciate the opportunity to clarify our methodology and rationale.
>
> 1. **Regarding Test Validity (Data Contamination and Context):**
> We agree that data contamination is a critical concern in modern benchmark design. Our data sourcing strategy was carefully designed to mitigate this risk. Sourcing from existing high-quality datasets is a common and efficient practice that allows for a focus on task design and annotation. Specifically:
>
> **Use of Test Splits**: For datasets like VGGSound, we exclusively sampled from its official test set. Similarly, Video-MME is itself a benchmark designed for evaluation. Standard practice dictates that these splits should not be included in the pre-training corpora of large models, thus minimizing the risk of data contamination.
>
> Novel Annotations, Not Reused Labels: For all sources, including FineVideo, **we did not use any of the original labels or captions**. Instead, we generated our entirely new set of annotations and QA pairs focused on synchronized reasoning. Therefore, even if a model has "seen" the raw video pixels, it has not been trained on our specific task or labels.
>
> **Addressing "Out of Context" Issues**: The concern about context is valid. This is precisely why rigorous human filtering is the final and most critical step in our pipeline. We built a Gradio UI where a human annotator views each video clip alongside its generated QA pair. **Any question that was ambiguous, unanswerable due to missing context from the original longer video, or otherwise flawed was explicitly rejected**. This ensures that every QA pair in the final benchmark is self-contained and valid.
>
> 2. **Regarding Raw Data Sourcing and Filtering**:
> We respectfully view our approach not as bypassing a challenge, but as building upon the strong, foundational work of the community.
> **Leveraging Mature, High-Quality Sources:** The datasets we chose, such as FineVideo, have already undergone their own extensive sourcing and filtering processes. As detailed in their own publications, they have robust mechanisms for curating high-quality, diverse video content. **These datasets are widely used and trusted by the community, and we believe their quality is well-established.**
>
> **Focusing on Our Core Contribution**: By leveraging these mature data sources, we were able to focus our efforts on our primary contributions: the design of a novel, synchronized reasoning task, the development of a scalable pipeline for complex QA generation, and the creation of the Daily-Omni Agent. We believe this is a more effective allocation of research effort than recreating the massive and complex undertaking of raw video collection from scratch.
> In summary, we are confident that our careful sourcing from test splits, creation of entirely novel annotations, and rigorous human filtering process ensure the validity and integrity of the Daily-Omni benchmark.
>
> ### Question 1: On Our Deliberate Choice of MCQ over Open-Ended Evaluation
> Thank you for this question.
> While we recognize the growing interest in open-ended formats, our decision to use multiple-choice questions (MCQ) was a deliberate one, driven by a commitment to creating a reproducible benchmark. We believe that for synchronized audio-visual reasoning, the evaluation methodology is as important as the data itself.
> Our primary concerns with the open-ended format are:
> - **Subjectivity and Bias in Scoring**: The evaluation of open-ended answers currently relies heavily on powerful external models like GPT-4o as judges. This introduces a significant confounding variable: the evaluation becomes a test not only of the model in question but also of the judge model's own biases, capabilities, and potential inconsistencies. This makes it difficult to draw firm, reproducible conclusions about a model's true performance.
> - **Conflation of Skills**: An open-ended format conflates the assessment of core multimodal reasoning with a model's text generation and stylistic abilities. A model might correctly understand the audio-visual context but fail to articulate it in a way the judge model prefers, and vice versa.
> Given that the primary aim of Daily-Omni is to isolate and specifically evaluate an MLLM's core multimodal reasoning capabilities, we firmly believe that the objectivity of the MCQ format is the most scientifically sound approach at this time.
>
> Once again, we sincerely thank you for your positive and constructive review. We have carefully considered all your points and believe that incorporating these clarifications will substantially strengthen the manuscript. Given that you have already rated our work as above the acceptance threshold, we hope our detailed responses have solidified your assessment and earned your strong support for our paper's acceptance.

---

### Official Review · Reviewer_CA6b · 2025-11-01

**Soundness:** 3
**Presentation:** 3
**Contribution:** 3
**Rating:** 6
**Confidence:** 4

**Summary:**

The submitted manuscript addresses the limitation of current multimodal large language models in synchronously processing cross-modal information from both visual and audio streams. To tackle this issue, the authors introduce a new benchmark dataset named Daily-Omni for Audio-Visual Questioning and Answering benchmark, along with a Daily-Omni QA Generation framework and a Daily-Omni-Agent model. The proposed system integrates open-source vision-language models, audio-language models, and ASR technology, enabling temporal-aware reasoning without the need for additional training. Overall, the work demonstrates a notable degree of novelty.

**Strengths:**

1. The manuscript clearly identifies the problem and presents a well-defined motivation.
2. The proposed Daily-Omni benchmark contributes to advancing research in audio-visual reasoning.
3. The Daily-Omni QA Generation framework shows strong potential for extensibility and future development.
4. The writing is clear and well-organized, making the paper easy to read and understand.

**Weaknesses:**

1.It is recommended to discuss the differences between this work and related studies such as *MMAU*, *AURA*, and *OmniVideoBench*.

2. The generalization experiments for Daily-Omni are insufficient. It remains unclear whether the model demonstrates a truly generalizable multimodal reasoning ability or if its effectiveness is limited to the constructed dataset.

3. The validity of the results may largely stem from the inherent capabilities of the large models used. How are the spatio-temporal associations between audio and visual elements verified?

4. Section 4.3 requires a more detailed explanation of aspects such as event pair matching, threshold selection, and grounding determination.

5. Considering that MUSIC-AVQA is one of the established benchmarks for audio-visual reasoning, why not conduct a more in-depth exploration on this dataset?

6. It is strongly recommended to use vector graphics for figures to improve visual quality and readability.

**Questions:**

My main questions are reflected in the Weaknesses Section.

---

> ### Author Response · Authors · 2025-11-18
>
> Thanks for your detailed review. We would like to address the weaknesses and questions you raised point by point.
> ### Weakness 1: On Comparison with Related Benchmarks
> Thank you for pointing out these highly relevant and important studies. We will add a detailed discussion of these works to our related work section in the final manuscript.
> Our analysis shows that while these are all valuable contributions, Daily-Omni addresses a distinct and complementary research gap:
> **MMAU** (Massive Multitask Audio Understanding): As a benchmark for massive multitask audio understanding, MMAU focuses exclusively on the audio modality for its multiple-choice QA tasks. In contrast, Daily-Omni is fundamentally **omni-modal**, designed to evaluate the complex reasoning that occurs at the intersection of both audio and visual streams.
> **OmniVideoBench**: This is a very recent and comprehensive benchmark. A key difference lies in the data creation process and scope. The questions in OmniVideoBench are manually annotated, which, while ensuring quality, lacks an **automated pipeline for scalability** like our Daily-Omni QA Generation Pipeline. Furthermore, it serves as a broad evaluation suite without proposing **a new benchmark model or agent** like our **Daily-Omni Agent**.
> **AURA**: This work is indeed conceptually similar to ours as it also proposes an automated pipeline for generating QA pairs. However, a critical distinction lies in the annotation process. AURA annotates audio and visual information separately, without a dedicated step for **aligning corresponding audio-visual events** (akin to our "Visual & Audio Event Alignment" step in Section 3.2). This separation means it may not effectively probe the fine-grained temporal reasoning and event synchronization that is the core focus of the Daily-Omni benchmark.
>
> In summary, these are all excellent and valuable studies. Our work complements them by specifically targeting the under-explored challenge of fine-grained, synchronized audio-visual reasoning in daily-life scenarios, supported by a scalable data generation pipeline and a strong, dedicated baseline agent. We appreciate you bringing them to our attention and will ensure they are properly discussed in the final version of our paper.
>
> ### Weakness 2: On the Generalization of Our Agent and Benchmark
> Thank you for raising this important point regarding the generalization capabilities of the Daily-Omni Agent. We agree that generalization is a critical aspect of evaluating any model.
> Our primary goal in designing the Daily-Omni Agent was to establish a strong performance baseline for omni-modal LLMs using powerful, off-the-shelf unimodal components. Regarding its generalization ability, we would like to offer two key clarifications:
>
> The Agent is a **Training-Free Framework**: It is crucial to note that the Daily-Omni Agent is a training-free architecture. We have not fine-tuned any of its components on Daily-Omni or any other specific dataset. Its performance is a direct reflection of the inherent capabilities of its underlying unimodal models (e.g., Qwen2.5-VL, Qwen2-Audio). These components have been pre-trained on massive, diverse datasets and possess strong, general-purpose audio and visual understanding abilities. **Therefore, the agent's capacity to understand novel audio-visual content is theoretically as broad as that of its foundational models.**
>
> The Benchmark is Sourced from Open-Domain Videos: The videos in the Daily-Omni benchmark itself are curated to be diverse and general. As detailed in **Section 3.1,** they are randomly sampled from multiple sources, which in turn contain a wide variety of YouTube videos spanning different eras, styles, and content types. This ensures our benchmark is **open-domain** and reflects the complexity of real-world scenarios, rather than being confined to a narrow, specific domain.
>
> In summary, given that our training-free agent is evaluated on an open-domain benchmark, we are confident that the observed performance is a strong indicator of its real-world multimodal reasoning capabilities, not an artifact of being narrowly tailored to a specific dataset.

---

> ### Author Response · Authors · 2025-11-18
>
> ### Weakness 3: On Isolating Multimodal Performance from LLM Capabilities
> This is a critical point, and we thank you for raising it. We agree that the performance of our agent stems from the power of its underlying models. However, our benchmark is specifically designed to isolate and measure the multimodal reasoning capability, ensuring that the results are not just a reflection of the language model's inherent knowledge or text-based reasoning power.
> We address this challenge in two main ways:
> Rigorous Quality Control During Data Creation: As detailed in **Section 3.2**, our "Quality Control" module is a crucial step in the data generation pipeline. We use multiple powerful, text-only LLMs to attempt to answer the generated questions without any audio-visual input. A**ny QA pair that can be consistently answered correctly through textual cues or common-sense reasoning alone is systematically filtered out and discarded.** This process acts as a strong safeguard against linguistic biases and ensures that a correct answer genuinely requires understanding the spatio-temporal audio-visual associations.
> Empirical Validation with Text-Only Models: We provide direct evidence for this in **Table 2** of our paper. **We evaluated several state-of-the-art text-only LLMs** (including Qwen2.5-Instruct 14B, Deepseek V3, and GPT-4o) on our benchmark. The results show that their performance is only marginally better than random guessing (25%). **This demonstrates that the language modality alone is insufficient to solve the tasks in Daily-Omni.**
>
> Therefore, we are confident that our benchmark effectively isolates and evaluates a model's purely multimodal reasoning abilities, factoring out the confounding variable of the LLM's text-based capabilities. The validity of the results stems from processing the spatio-temporal information, which is precisely what we aim to measure.
>
> ### Weakness 4: On Providing Further Details for Section 4.3
> Thank you for this constructive feedback. We agree completely that Section 4.3 would benefit from a more detailed technical description of our alignment process. We will expand this section significantly in the final version.
> Here is a step-by-step breakdown of the "Smart Alignment" methodology, addressing the aspects you raised:
>
> **Initial Comprehensive Captioning**: First, we use Qwen2.5-VL and Qwen2-Audio to generate a complete set of audio and visual event captions for the entire video.
> Event Pair Matching (Visual Event Filtering): The initial captions and the user's question are fed into our reasoning LLM (Qwen2.5-Instruct-14B). The LLM acts as a semantic filter, outputting a list of visual events from the captions that it deems relevant for answering the question.
>
> **Grounding Determination** (Visual to Audio): For each relevant visual event in this list, we then perform temporal grounding.
> We prompt Qwen2.5-VL with the visual event description (e.g., "throwing a ball") and ask it to identify the specific time segment in the video where this event occurs (e.g., 5-10s).
> This uni-directional grounding approach (from visual to audio) was a deliberate design choice. We found in our experiments that current models are significantly more proficient at visual event grounding than audio event grounding, which tends to be less precise.
>
> **Audio Alignment and Final Pairing**:
> We take the grounded time segment (e.g., 5-10s) and crop the corresponding audio stream.
> This short audio clip is then fed to Qwen2-Audio. We ask it to confirm which of the initial, full-video audio captions (e.g., "laughter heard") are present within this specific clip.
> If an audio event is confirmed, we create the final aligned pair: (`throwing a ball` at 5-10s, `laughter heard` at 5-10s).
>
> **Threshold Selection**: The duration threshold you noted is used to filter the grounded visual segments. We only proceed with audio alignment if the grounded segment's duration is below an empirically set threshold: 6 seconds for 30s videos and 7 seconds for 60s videos. This prevents noise and ambiguity from overly long or poorly grounded segments.
>
> We will integrate these technical details into Section 4.3 to provide a clear and reproducible account of our methodology. Furthermore, to ensure full transparency, we will make our code publicly available, which will include the complete pipeline for the Daily-Omni Agent. Thank you again for helping us strengthen the paper.

---

> ### Author Response · Authors · 2025-11-18
>
> ### Weakness 5: On the Rationale for a New Benchmark Beyond MUSIC-AVQA
>
> Thank you for this excellent question. We agree that MUSIC-AVQA is a foundational and highly influential benchmark in the audio-visual reasoning field. Our decision to create a new benchmark, rather than extending our work on MUSIC-AVQA, was driven by our specific research goal to address a different type of challenge.
> The primary distinction lies in the domain and the nature of the audio-visual interaction:
>
> **Domain Focus**: MUSIC-AVQA is expertly focused on structured music performance scenarios. While visually grounded, the reasoning often centers on the audio modality—identifying instruments, their sounds, and their interactions.
> **Content Diversity**: Our goal with Daily-Omni was to evaluate reasoning in **open-domain**, unstructured, daily-life contexts. We specifically aimed for a benchmark with richer and more dynamic content on both sides. The videos are more varied and complex, and the audio includes a wide array of sounds beyond music, such as human speech, ambient sounds, and distinct sound events, which require a different, often more causal, type of reasoning.
>
> That being said, we want to reassure you that **our benchmark does not ignore the music performance domain**. Our data is sourced from broad, diverse datasets like **VGGSound** and **FineVideo**, which **naturally include clips of musical performances** alongside countless other daily-life events. Therefore, the capabilities tested in MUSIC-AVQA are a subset of the broader, more varied reasoning skills evaluated in Daily-Omni. We believe that by creating this new benchmark, we are pushing the community to develop models that are robust not just in specific, structured domains, but in the complex scenarios of everyday life.
>
> ### Weakness 6: On Improving Figure Quality with Vector Graphics
> Thank you for your constructive feedback regarding the figures. We will ensure all figures are prepared in a high-resolution vector format for the camera-ready version to enhance readability and visual quality. We would be very grateful if you could point out any specific figures that you felt particularly needed improvement, as this would help guide our revisions.
>
>
> We have carefully considered all your points and believe that incorporating these detailed clarifications and promised revisions will substantially strengthen the manuscript. We hope our responses have fully addressed your concerns and further highlighted the value of our contributions. With this in mind, we would be very grateful if you would reconsider your evaluation of our work.

---

> > ### Comment · Reviewer_CA6b · 2025-11-28
> >
> > Thank you very much for the authors’ detailed response, which addressed the vast majority of my concerns. I will maintain my positive review score.

---

### Public Comment · ~Yuewen_Zhu1 · 2025-11-13
**Inquiry Regarding Qwen 2.5 Omni Results on the Leaderboard and Our Reproduction Data**

Hello, thank you to the author team for your valuable work and for sharing it with the community.

While studying your research, we were intrigued by the results on your leaderboard and decided to reproduce the performance of Qwen 2.5 Omni using your provided data. However, our reproduction results show a significant discrepancy compared to your published data. It appears that the reported performance for Qwen 2.5 Omni has been severely underestimated.

According to our detailed evaluation, the actual performance of Qwen 2.5 Omni 7B is exceptional, with its overall score even surpassing that of your proposed Daily-Omni Agent.

Here are the specific scores we obtained for Qwen 2.5 Omni 7B:

  "Event Sequence": 57.19,

  "AV Event Alignment": 50.42,

  "Inference": 77.27,

  "Reasoning": 78.29,

  "Context understanding": 56.48,

  "Comparative": 70.99,

  "30s Subset": 63.68,

  "60s Subset": 62.0,

  **"Avg": 62.91**

This result (Avg: 62.91) exceeds that of your proposed Daily-Omni Agent. Therefore, we have reservations about the reliability of your experimental data and believe that, from a pure performance perspective, the claimed significance of Daily-Omni over existing mainstream open-source models requires further validation.

We sincerely suggest that you review your evaluation pipeline for Qwen 2.5 Omni to verify if there were any issues with configuration or inference, and update the leaderboard with accurate results. **It is worth noting that we raised this issue in your official GitHub repository on July 17th but have not yet received a response.**

We look forward to your response and corrections.

---

> ### Author Response · Authors · 2025-11-14
> **Qwen2.5-Omni Evaluation Issue**
>
> Thank you for your interest in Daily-Omni. In response to the questions you raised, we have carefully examined the evaluation process of **Qwen2.5-Omni**.
>
> We set up the testing environment according to instructions in an early commit `655410969153857be50300566faf567bf30cde32` (April 8). We use `Qwen2_5OmniModel` and `Qwen2_5OmniProcessor` for AQVA task as instructed in [`cookbooks/video_information_extracting.ipynb`](https://github.com/QwenLM/Qwen2.5-Omni/blob/655410969153857be50300566faf567bf30cde32/cookbooks/video_information_extracting.ipynb).
> Using neither `You are a helpful assistant.` or `You are Qwen, a virtual human developed by the Qwen Team, Alibaba Group, capable of perceiving auditory and visual inputs, as well as generating text and speech.` **produce results consistent with those reported in our paper**. We have provided the full evaluation code and further details in the corresponding GitHub issue.

---

### Author Response · Authors · 2025-11-27
**Gentle Reminder regarding Author Response**

Dear Reviewers,

We sincerely appreciate the time and effort you have dedicated to reviewing our paper.
We posted our responses 8 days ago, **providing detailed explanations and clarifications for all the weaknesses and questions raised in your reviews**.
As the discussion period is progressing, we would be grateful for your feedback on whether our responses have satisfactorily resolved your concerns. We remain available to answer any further questions you may have.

Best regards,

The Authors

---

### Author Response · Authors · 2025-12-03
**Summary of the discussion period**

Dear Area Chair,
We have provided comprehensive responses to address the concerns and questions raised in all four reviews. We are encouraged that Reviewer YCPb (initially 2) indicated an inclination to raise their score, while Reviewer CA6b (initially 6) maintained their positive assessment following our rebuttal.

Regrettably, Reviewers 1BVk (initially 6) and tGax (initially 4) did not respond before the discussion period closed due to the leaked reviewer information. Their primary concerns centered on comparisons with related work. in our rebuttal, we provided detailed comparisons and discussions regarding all the specific studies they mentioned.
We are confident that these clarifications effectively resolve their concerns and believe that, had the discussion continued, their ratings would likely have improved. **Consequently, we believe the fair evaluation of our work corresponds to an average rating of 5.5 or higher.**

**We kindly request that you review our responses, particularly those addressing Reviewers 1BVk and tGax, to fully evaluate the contribution and novelty of our work.**

Sincerely,
The Authors

---

### Meta-Review · Area_Chair_L7Kc · 2026-01-06

**Summary:**

This paper introduces Daily-Omni, a new audio-visual question answering benchmark targeting synchronous cross-modal reasoning in everyday scenarios, together with an automated QA generation pipeline and a training-free baseline agent. Despite the existence of prior audio-visual benchmarks such as AVUT and WorldSense, Daily-Omni remains a valuable contribution due to its focus on fine-grained daily-life scenarios, diverse task coverage, and an efficient, scalable benchmark construction pipeline.

However, a major concern remains unresolved regarding the reported experimental results. Specifically, the performance numbers on Daily-Omni appear unexpectedly low and inconsistent with independently verified evaluations. For example, Qwen 2.5 Omni 7B achieves approximately 62.91% accuracy when tested externally, rather than the 47.45% reported in Table 2 of the paper. Similar discrepancies have been reported and verified by other sources.

As a consequence, the reported performance of the proposed Daily-Omni-Agent (61.82%) no longer represents a meaningful baseline advantage and appears to lag behind existing models under corrected evaluations. This directly calls into question the strength and validity of the paper’s third contribution (about developing Daily-Omni-Agent). Clarifying the evaluation protocol and reconciling these discrepancies are essential for establishing the reliability and impact of the benchmark and its associated baseline.

In addition, the author's response (to Reviewer YCPb-Q1) regarding the maximum video lengths that recent video understanding LLMs can process is not accurate, as many contemporary models are now capable of handling hour-long videos effectively.

**Reviewer Concerns:**

The concerns raised by Reviewer 1BVk (W2), Reviewer YCPb (Q1, W2), and Reviewer tGax (Q3) remain unresolved due to the presence of experimentally unverifiable and underestimated results, as well as an underestimation of the current capability of video understanding LLMs to process long videos. Most other concerns have been adequately addressed.

**Reviewer Scores:**

I expect Reviewer CA6b to maintain or possibly increase their rating, while the other reviewers are likely to maintain their current ratings due to the remaining unresolved concerns.

---

### Decision · Program_Chairs · 2026-01-26

Reject